# Identification of Task Affinity for Multi-Task Learning based on Divergence of Task Data

## Abstract

Multi-task learning (MTL) can significantly improve performance by training shared models for related tasks. However, due to the risk of negative transfer between mismatched tasks, the effectiveness of MTL hinges on identifying which tasks should be learned together. In this paper, we show that for tabular datasets, this affinity between a pair of tasks can be predicted based on static features that characterize the relationship between the datasets of these tasks. Specifically, we show that we can train a regression model for predicting pairwise task affinity based on computationally efficient features, requiring ground-truth affinity values for only a small, random sample of task pairs to generalize across all possible pairs. We demonstrate on three benchmark tabular datasets that our proposed approach can predict affinity more accurately at lower computational cost than existing methods for identifying task affinity, which treat task data as black boxes and require training-based signals. Our work provides a practical and scalable solution to task grouping for MTL, enabling its effective application to tabular datasets with large numbers of tasks.

## 1 Introduction

Multi-task learning (MTL) aims to improve generalization by exploiting shared structure across similar tasks. However, its success critically depends on *which* tasks are learned together, as poorly chosen groups of tasks can lead to negative transfer. Despite the broad use of MTL across domains, principled methods for selecting task groups remain underexplored. The core challenge reduces to identifying *affinity* between tasks: how likely a given pair of tasks are to benefit from joint training.

Recent efforts have sought to open the black box of MTL and characterize task relationships. Data-driven methods such as Linear Surrogate (Li et al., 2023), GRAD-TAE (Li et al., 2024), and MT-GNet (Song et al., 2022) learn predictors of MTL performance for groups, but they require expensive ground-truth supervision from training large numbers of MTL models. HOA (Standley et al., 2020) exhaustively measures MTL performance for all pairs of tasks, while TAG (Fifty et al., 2021) estimates affinities by repeatedly probing gradients in a single all-task MTL model. Although insightful and effective, these approaches are computationally demanding, often requiring joint training across many task groups to generate supervision labels, and they scale poorly to large sets of tasks.

Other lines of work focus on identifying the characteristics of datasets that correlate with task affinity, e.g., distributional similarity (Ben-David & Borbely, 2008) or single-task predictors of MTL affinity (Bingel & Søgaard, 2017). However, these methods stop short of directly predicting MTL performance and do not provide scalable mechanisms for grouping large sets of tasks.

In this paper, we propose a practical and scalable framework for predicting *pairwise MTL performance* using features that capture the structural and statistical relationships between tabular tasks. We construct pairwise feature vectors from tabular dataset statistics, input-space geometry, and representation similarity—many of which can be precomputed per task and reused across pairings—substantially reducing computational cost. We then train a regression model on a small, random sample of task pairs with ground-truth MTL gains—often orders of magnitude lower than required by prior methods—that generalizes to all remaining pairs without exhaustive joint training.

Our approach avoids the combinatorial cost of training on all task pairs, enables scalable and inexpensive estimation of pairwise affinities even for large sets of tasks, and provides interpretable,

data-driven predictions that can be integrated into downstream task-grouping methods. We validate our approach on three widely used tabular MTL benchmarks, demonstrating that it accurately predicts task-pair affinity with only a small number of training pairs. Furthermore, we show that these predictions can guide task-grouping methods to efficiently identify high-performing task groups, providing a lightweight and practical alternative to resource-intensive prior approaches.

## 2 PROBLEM FORMULATION

We consider a set of $n$ tasks, denoted by $\mathcal{T} = \{t_1, t_2, \ldots, t_n\}$. Each task $t \in \mathcal{T}$ is associated with a dataset $D_t = \{(x_t^{(i)}, y_t^{(i)})\}_{i=1}^{|D_t|}$, where $x_t^{(i)} \in \mathbb{R}^p$ and $y_t^{(i)}$ denote the input vector and target label, respectively. We assume a common input dimension $p$ across all tasks. For a model with parameters $\theta$, the prediction performance on task $t$ is measured by a loss function:

$$L_t(D_t; \theta) = \frac{1}{|D_t|} \sum_{i=1}^{|D_t|} \ell_t\big(\hat{y}_t(x_t^{(i)}; \theta), y_t^{(i)}\big),$$

where $\ell_t(\cdot, \cdot)$ is the task-specific positive loss (e.g., MSE or cross-entropy) and $\hat{y}_t(\cdot; \theta)$ denotes model predictions for task $t$. A task group is defined as a subset $G \subseteq \mathcal{T}$, which may contain between $1$ and $n$ tasks. A multi-task learning (MTL) algorithm jointly trains a model with shared parameters $\theta_{\text{MTL}}^G$ across all tasks in $G$ by minimizing the average training loss:

$$\theta_{\text{MTL}}^G = \arg\min_\theta \frac{1}{|G|} \sum_{t \in G} L_t(D_t; \theta). \tag{1}$$

Similarly, $\theta_{\text{STL}}^t$ denotes the parameters of the single-task model (STL) trained independently for $t$:

$$\theta_{\text{STL}}^t = \arg\min_\theta L_t(D_t; \theta). \tag{2}$$

**Task Affinity** We measure *task affinity* in terms of **MTL gain**: the relative improvement in a task's loss when trained jointly with another task compared to training it independently. For a task pair $G = \{t_i, t_j\}$ with a trained MTL model $\theta_{\text{MTL}}^{\{t_i, t_j\}}$, the observed MTL gain for $t_i$ is

$$\texttt{gain}_{t_j \to t_i} = \frac{L_{t_i}(D_{t_i}; \theta_{\text{STL}}^{t_i}) - L_{t_i}(D_{t_i}; \theta_{\text{MTL}}^{\{t_i, t_j\}})}{L_{t_i}(D_{t_i}; \theta_{\text{STL}}^{t_i})}. \tag{3}$$

A positive value indicates that joint training with $t_j$ improves $t_i$'s performance (positive transfer), while a negative value indicates degradation (negative transfer). This definition naturally generalizes to larger groups by replacing $\{t_i, t_j\}$ with $G$.

**Objective: Predicting Pairwise MTL Gains** Our goal is to predict **pairwise task-affinity** (i.e., **MTL gains**) using statistical features that characterize the datasets of the tasks and their relationships, without performing exhaustive joint training over all task pairs. Formally, for any pair of tasks $\{t_i, t_j\}$ with datasets $D_{t_i}$ and $D_{t_j}$, we seek to estimate gains $\texttt{gain}_{t_j \to t_i}$ and $\texttt{gain}_{t_i \to t_j}$, as defined in Equation (3), which quantify the relative improvement (or degradation) in one task's loss when trained jointly with the other. We assume access to ground-truth gains for a small subset of task pairs $\mathcal{G}_{\text{train}} \subset \binom{\mathcal{T}}{2}$, for which we perform joint MTL training. The objective is to learn a predictive model that generalizes to unseen pairs in $\mathcal{G}_{\text{test}} = \binom{\mathcal{T}}{2} \setminus \mathcal{G}_{\text{train}}$, thereby enabling scalable estimation of pairwise MTL gains.

## 3 AFFINITY PREDICTION WITH TASK-RELATION FEATURES

Exhaustively training MTL models for all $\binom{n}{2}$ task pairs is computationally prohibitive. So, our goal is to understand whether the *pairwise MTL gain*, as defined in Equation (3), can be predicted based on statistical features that characterize the relationships between the datasets of tasks $t_i$ and $t_j$. Our approach proceeds in two stages: (i) design of task-relation features $\phi_{i,j}$ that capture structural and statistical relationships between $D_{t_i}$ and $D_{t_j}$, and (ii) supervised learning of a regression function

$f(\phi_{i,j};\omega)$ that maps these features to estimated gains. This design ensures computational tractability: rather than performing MTL training for every possible pair of tasks (which scales quadratically with the number of tasks), we collect ground-truth gains only for a small subset of task pairs and rely on a predictive model to generalize to unseen pairs. To capture relationships between datasets of tasks, we extract a diverse set of features quantifying statistical and structural differences, including relative dataset sizes and various distributional distances. Each feature vector $\phi_{i,j} = \phi(D_{t_i}, D_{t_j})$ is thus computable from the task datasets directly, without performing any MTL training. The predictive function $f(\phi_{i,j};\omega)$ is trained on a limited set of ground-truth pairwise gains $\mathcal{G}_{\text{train}}$ and can efficiently estimate gains for all remaining task pairs in $\mathcal{G}_{\text{test}} = \binom{\mathcal{T}}{2} \setminus \mathcal{G}_{\text{train}}$.

While prior work has explored task affinities in transfer learning and MTL (Bingel & Søgaard, 2017; Standley et al., 2020), these studies either rely on task-specific heuristics or require repeated model evaluations across task subsets, limiting scalability. In contrast, our approach systematically considers **a broader set of features between task pairs** that capture various aspects of dataset similarity and enable **scalable prediction of pairwise MTL gains** without exhaustive training, making it applicable to larger sets of tasks.

**Hypothesis** We hypothesize that tasks with more similar data distributions are more likely to yield positive gains under joint training. The intuition is that similar input patterns imply shared underlying structures or decision boundaries, making shared representation learning more effective. To capture this, we leverage both statistical and structural features—such as dataset size and class imbalance—to quantify task relatedness. Our central insight is that **if two tasks appear similar in terms of their data distribution, they are more likely to benefit from being learned together.**

### 3.1 Designing Task-Relation Features

To enable predictive modeling of MTL gains, we construct a comprehensive set of features that quantify the relationship between pairs of tasks using their datasets. As usual, we assume that for each task, the training and test sets are drawn from the same distribution. When computing task-relation features, we use both the training and test data to capture the overall structure of the task. For features that describe the label distribution, we compute them using only the training data to avoid leakage. Below, we describe the categories of task-relation features used in our study.

**Features based on Dataset Size** Basic structural information about the datasets provides a first-order characterization of task similarity. We record the **total size** of the combined dataset as well as the **relative size** of each task:

$$\texttt{Data-Size} = |D_{t_i}| + |D_{t_j}|, \quad \texttt{Data-Ratio} = \frac{|D_t|}{|D_{t_i}| + |D_{t_j}|}, \quad t \in \{t_i, t_j\}. \tag{4}$$

To account for imbalance, we also compute the normalized difference in size, which reflects whether one dataset dominates the MTL training:

$$\texttt{Size-Gap} = \frac{||D_{t_i}| - |D_{t_j}||}{|D_{t_i}| + |D_{t_j}|}. \tag{5}$$

**Distance- and Distribution-Based Features** To quantify structural and distributional differences between tasks, we compute several statistics based on pairwise distances between samples. For each task $t$, we define the **average intra-task pairwise distance**:

$$d_t = \mathbb{E}_{x,x'\sim D_t}\big[\|x - x'\|\big], \tag{6}$$

where $x, x' \in \mathbb{R}^p$ are two independent samples drawn from $D_t$ and $\|\cdot\|$ denotes the standard Euclidean (L2) norm in the $p$-dimensional input space. For large datasets, $d_t$ is estimated from a random subset of sample pairs to stay computationally tractable. Given two tasks $t_i$ and $t_j$, we compute the **cross-task distance**, measuring how far samples from the two tasks lie from each other:

$$d_{(t_i+t_j)} = \mathbb{E}_{x\sim D_{t_i}, x'\sim D_{t_j}}\big[\|x - x'\|\big]. \tag{7}$$

Normalized versions of these distance measures (e.g., $\frac{|d_{t_i} - d_{t_j}|}{d_{t_i} + d_{t_j}}$ or $\frac{d_{(t_i+t_j)}}{\sqrt{d_{t_i}\, d_{t_j}}}$) provide scale-invariant indicators of similarity in these metrics. We also compute the **energy distance**, a statistical measure

of dissimilarity between two input distributions. It is based on pairwise distances among samples across and within the two datasets. Energy distance between the input distributions of $t_i$ and $t_j$ is:

$$\texttt{Energy-Distance} = 2d_{(t_i+t_j)} - d_{t_i} - d_{t_j}, \tag{8}$$

where larger values indicate greater dissimilarity between input distributions. To capture differences in central tendency, we also compute the **euclidean distance between feature-wise means**:

$$\texttt{Feature-Mean Gap} = \|\mu_{t_i} - \mu_{t_j}\|, \tag{9}$$

where $\mu_t = \mathbb{E}_{x \sim D_t}[\mathbf{x}]$ is the mean feature vector for task $t$. This metric reflects whether the datasets are centered around similar regions in the input space. Larger distances suggest different average behavior in the input features. Together, these features summarize both the spread (variance), relative positioning, and overall divergence of the two datasets in input space.

**Representation-Based Features** We further assess task similarity in terms of their raw or transformed feature representations. In contrast to the distance-based features described earlier, which rely on Euclidean norms, here we replace the distance metric with an angular measure of alignment. Specifically, we compute the average **cosine similarity** between samples from two tasks:

$$\texttt{Cosine-Sim} = \mathbb{E}_{x \sim D_{t_i}, \, x' \sim D_{t_j}} \left[ \frac{\langle x, x' \rangle}{\|x\| \cdot \|x'\|} \right]. \tag{10}$$

Here $x$ and $x'$ denote samples from $D_{t_i}$ and $D_{t_j}$, respectively. **Principal component analysis (PCA)** provides another view: we compute the top-$k$ principal components for each $D_t$ and measure their alignment by averaging the absolute cosine similarity of the corresponding components:

$$\texttt{PCA-Align} = \frac{1}{k} \sum_{\ell=1}^{k} |\langle u_\ell^{(i)}, u_\ell^{(j)} \rangle|, \tag{11}$$

where $u_\ell^{(i)}$ and $u_\ell^{(j)}$ are the $\ell$-th principal components of $D_{t_i}$ and $D_{t_j}$, respectively. The absolute value $|.|$ removes the sign ambiguity of PCA directions (since principal components are defined only up to a sign flip), making the measure invariant to direction reversals. We also consider a **rank-based divergence** measure, where for each feature $k \in \{1, \ldots, p\}$, we pool the values from both tasks, assign ranks, and compute the difference in mean ranks between tasks. The overall divergence is the average absolute rank gap across features:

$$\texttt{Rank-Div} = \frac{1}{p} \sum_{k=1}^{p} |\bar{r}_{t_i}^{(k)} - \bar{r}_{t_j}^{(k)}|, \tag{12}$$

where $\bar{r}_t^{(k)}$ denotes the mean rank of feature $k$ for task $t$. The absolute value $|.|$ captures the magnitude of divergence, ignoring which task has higher or lower average rank. This metric reflects whether the relative ordering of samples differs systematically between tasks.

**Graph-Based Features** We quantify how much the two datasets are interwoven in feature space by measuring their topological overlap. To measure this, first we construct a $k$-nearest neighbor (KNN) graph on the combined data $D_{t_i} \cup D_{t_j}$, where each point is connected to its $k$ closest points (in Euclidean distance, with $k$ chosen as a small constant such as 5 or 10). We then compute the **cross-link ratio**, the fraction of edges that connect samples from different tasks, where a higher ratio indicates stronger intermingling of the two datasets in feature space:

$$\texttt{Cross-Link} = \frac{\text{Number of cross-task edges}}{\text{Total KNN edges}}. \tag{13}$$

### 3.2 Predictive Modeling of Pairwise MTL Gains

Given access to ground-truth MTL gains for a subset of task pairs $\mathcal{G}_{\text{train}}$, we adopt a data-driven approach to predict pairwise MTL gains using the features introduced in the previous section. For each task pair, we first compute a pairwise feature vector $\phi_{i,j}$, which encodes the relationship between the datasets of the two tasks. Our goal is to learn a regression model that predicts MTL gain based on these task-pair features, providing an efficient alternative to exhaustively training for all task pairs.

**Polynomial Regression**  To capture higher-order interactions between features, we expand each feature vector using second-degree polynomial terms, denoted $\Phi(\phi_{i,j})$. We then fit a regularized regression model that minimizes the following objective over all labeled training pairs:

$$\min_{\omega} \sum_{\{t_i,t_j\}\in\mathcal{G}_{\text{train}}} \left(\text{gain}_{t_i\to t_j} - \omega^\top\Phi(\phi_{i,j})\right)^2 + \lambda\|\omega\|_2^2, \tag{14}$$

where $\text{gain}_{t_i\to t_j}$ denotes the ground-truth MTL gain for task $t_j$ when trained with $t_i$, $\Phi(\phi_{i,j})$ is the quadratic feature expansion, and $\lambda$ is the $\ell_2$ regularization coefficient. While a linear variant provides a simpler and more interpretable baseline, the quadratic model better captures feature interactions and consistently achieves higher prediction accuracy (Appendix A.3). We therefore adopt the quadratic model as our default predictor.

## 4 Experiments and Results

We evaluate our approach on three benchmark MTL datasets. We begin with an analysis of the proposed task-relation features, and then provide a comparison to baseline prediction approaches.

### 4.1 Experimental Framework

**Benchmark Datasets**  We demonstrate our approach on three benchmark tabular datasets, which have been widely used in multi-task learning studies (Zhang & Yang, 2021). Table 1 summarizes these benchmark datasets, with additional details in Appendix A.1. **School** (Bakker & Heskes, 2003; Han & Zhang, 2015) involves predicting student exam scores across 139 schools, each treated as a separate task, using both school- and student-level features. **Chemical (MHC-I)** (Jacob et al., 2008; Zhou & Zhao, 2015) involves predicting peptide–molecule binding affinities for 35 molecules, each treated as a task. **Landmine** (Xue et al., 2007; Jawanpuria et al., 2015) contains data from 29 landmine fields, each treated as a task, with the goal of classifying whether a sample corresponds to a landmine. In all three benchmarks, each task has its own distinct set of data. We provide descriptive statistics for each benchmark dataset in Appendix A.1, which help to contextualize differences in task relationships and MTL performance across the three benchmarks.

Table 1: Benchmark tabular datasets and their characteristics.

| Dataset | Task Type | Loss Metric | #Tasks | Dataset Size ($|D_t|$) | | |
| --- | --- | --- | --- | --- | --- | --- |
| | | | | Avg. | Min. | Max. |
| **School** | Regression | Mean Squared Error (MSE) | $|\mathcal{T}| = 139$ | 110 | 22 | 251 |
| **Chemical** | Classification | Binary Cross-Entropy | $|\mathcal{T}| = 35$ | 435 | 22 | 2,368 |
| **Landmine** | Classification | Binary Cross-Entropy | $|\mathcal{T}| = 29$ | 511 | 445 | 690 |

**MTL Model Architecture and Hyperparameter Search**  For all three benchmarks, we employ feed-forward neural networks for both STL and MTL. Note that the design of the MTL architectures is not a vital component of our contribution. In fact, *our goal is orthogonal to the performance of specific MTL architectures and methods*, and our pairwise MTL gain prediction approach can be applied to and potentially benefit a wide range of MTL methods. For each benchmark, we perform a separate hyperparameter search on selected task groups to identify suitable architectures. STL models are trained first to establish task baselines, followed by MTL training on all task pairs $\binom{|\mathcal{T}|}{2}$ (9591 pairs for *School*, 595 for *Chemical*, 406 for *Landmine*) to obtain ground-truth pairwise gains. For training the gain predictor, we randomly sample 10–50% of task pairs as the labeled training set and reserve the remaining $> 50\%$ for testing, ensuring evaluation on unseen task pairs. Finally, we train higher-order task groups to assess the effectiveness of predicting gains for group selection.

### 4.2 Analysis of Task-Relation Features

A central question in our framework is which task-relation features are most informative for predicting pairwise MTL gains. To investigate this, we first evaluate the predictive power of individual features, quantifying how well each correlates with observed MTL gains. This highlights both strong

predictors and those that are redundant or noisy. Based on these results, we then construct a compact feature subset that balances predictive power with interpretability and avoids collinearity. The selected subset forms the basis for all subsequent modeling experiments.

### 4.2.1 PREDICTIVE POWER OF INDIVIDUAL FEATURES

We train a quadratic regression model using each feature in isolation and measure $R^2$ between the predicted and observed MTL gains. We also provide a detailed analysis of the correlation between features and MTL gains in Appendix A.2.2; here, we provide results based only on predictive performance $R^2$. Size-related features (`Data-Size`, `Data-Ratio`, `Size-Gap`) exhibit strong predictive power, particularly for *Chemical* and *Landmine*, highlighting the importance of data availability and balance between tasks. Distributional measures such as `Energy Distance` and `Feature-Mean Gap` consistently provide useful signal across all benchmarks, underscoring the crucial role of similarity between data distributions. Representation-based features (e.g., `Cosine Similarity`, `PCA Alignment`) are weak predictors on their own, though `Rank Divergence` achieves moderate utility in *Chemical* and *Landmine*. Finally, graph-derived feature `Cross-Link` exhibits strong predictive power in *Landmine* but inconsistent elsewhere.

Figure 1: Predictive power of each task-relation feature in isolation, based on quadratic regression of pairwise MTL gain. Each bar represents the $R^2$ *value between the predicted and true value of pairwise MTL gain.*

While predictive power varies across the benchmarks, features reflecting dataset size imbalance and distributional divergence are consistently informative. This suggests that our predictors can serve as lightweight estimators of MTL affinity, especially when combined into a broader feature set.

### 4.2.2 SELECTED SUBSET OF FEATURES

Guided by these correlation and prediction results, we curate a compact subset of task-relation features, which balances interpretability and predictive accuracy. The final selection includes: `Data-Size`, `Data-Ratio`, `Normalized Distances` ($|d_{t_i} - d_{t_j}| \div \sqrt{d_{t_i} \cdot d_{t_j}}$, $d_{(t_i+t_j)} \div \sqrt{d_{t_i} \cdot d_{t_j}}$), `Energy Distance`, `Feature-Mean Gap`, and `Rank Divergence`. These features span key axes of task-pair variation: *data availability* and *balance*, *distance-based similarity*, *distributional divergence*, and *structural alignment*. We excluded features that are redundant or have weak predictive power (e.g., `PCA-Align`, `Cosine-Sim`, `Cross-Link`). Our selected subset of features provides a compact but informative basis for regression modeling of MTL gains.

### 4.3 ANALYSIS OF PREDICTION PERFORMANCE

We evaluate our feature-based prediction approach in terms of both predictive accuracy and computational efficiency, comparing it to recent baselines for estimating pairwise task affinities. **TAG** (Fifty et al., 2021) estimates affinities from a single MTL model by calculating inter-task influence scores, requiring $n$ additional forward and backward passes per update. We replicate this by training one joint model per dataset and aggregating influence scores across the training iterations. **GRAD-TAE** (Li et al., 2024) projects task gradients from a baseline MTL run and fits logistic regressions

on randomly sampled task subsets. We reproduce this by sampling 2,000 task subsets, estimating their performance, and defining affinity between task pair $\{t_i, t_j\}$ based on the average loss across subsets where both tasks co-occur. We evaluate both baselines in terms of correlation with ground-truth pairwise MTL gains. We provide comparisons with additional baselines in Appendix A.5.

### 4.3.1 PREDICTIVE ACCURACY

We first compare the predictive accuracy of our feature-based approach against TAG and GRAD-TAE, across the *School*, *Chemical*, and *Landmine* benchmarks. As shown in Table 2, our models consistently achieve higher correlation between predicted affinities and actual MTL gains across all benchmarks. On *School*, which is particularly challenging to predict, our approach clearly outperforms the baselines. On *Chemical*, our approach achieves 0.50 correlation, substantially outperforming GRAD-TAE (0.15). On *Landmine*, our approach reaches 0.58, well above TAG (0.34) and GRAD-TAE (−0.26). These results highlight the effectiveness of our feature-based prediction over training-intensive baselines.

Table 2: Comparison between methods for pairwise MTL gain predictions in terms of predictive accuracy, measured as correlation between predicted pairwise affinity and actual MTL gain for each benchmark dataset.

| | Correlation between Prediction and Actual MTL Gain | | |
| | **Inter-Task Affinity** | **GRAD-TAE** | **Feature-Based** |
| **Dataset** | (Fifty et al. (2021)) | (Li et al. (2024)) | Quadratic (Ours.) |
|---|---|---|---|
| **School** | +0.002 ± 0.00 | -0.002 ± 0.00 | **+0.13± 0.02** |
| **Chemical** | +0.06 ± 0.04 | +0.15± 0.00 | **+0.50± 0.03** |
| **Landmine** | +0.34 ± 0.02 | -0.26± 0.00 | **+0.58± 0.03** |

### 4.3.2 COMPUTATIONAL EFFICIENCY

We next compare runtime efficiency across the same set of baselines. **TAG** requires one complete MTL training plus $n$ extra forward and backward passes per gradient update to compute the $n \times n$ affinity matrix. **GRAD-TAE** requires $M$ complete MTL trainings, $O(n)$ gradient computations and storage, and $m$ logistic regressions on sampled subsets. This scaling incurs substantial overhead: increasing $M$ from 1 to 5 nearly quintuples runtime with only marginal accuracy gains.

Our approach avoids this bottleneck. After collecting ground-truth MTL gains for a small subset of task pairs, the remaining steps—computing pairwise statistics, training a lightweight predictor, and inferring affinities for all other pairs—are efficient and scale gracefully. Crucially, while the initial collection of ground truths does incur substantial cost, it scales only with the number of pairs sampled. As shown in Figure 2, increasing the number of training pairs steadily improves prediction accuracy ($R^2$), while runtime grows at a much slower rate compared to GRAD-TAE.

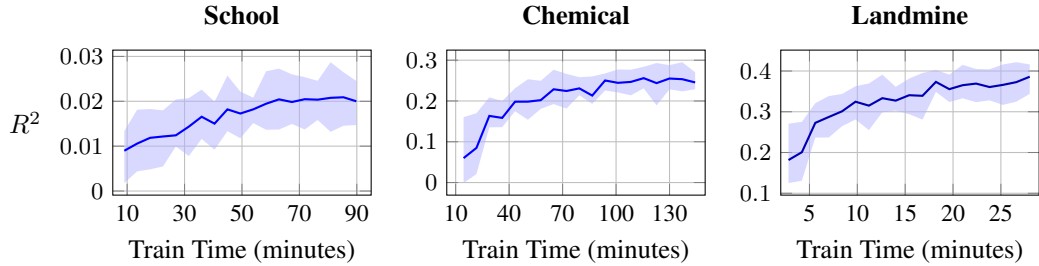

Figure 2: Prediction performance ($R^2$) of our approach vs. training time, with shaded bands indicating the interquartile range (25th–75th percentile). Training set $\mathcal{G}_{\text{train}}$ comprises 10–50% of all task pairs.

As shown in Figure 3, this design yields the best trade-off between runtime and accuracy. With using only 25% of all pairs as training data, our method achieves 0.13 correlation in 18 minutes on *School*, outperforming TAG (89 minutes, −0.01 correlation) and GRAD-TAE (8–39 minutes, negative correlation); on *Chemical*, we attain 0.50 in 29 minutes vs. GRAD-TAE over 91 minutes for weaker accuracy; on *Landmine*, we attain 0.58 in 5.6 minutes, outperforming both baselines.

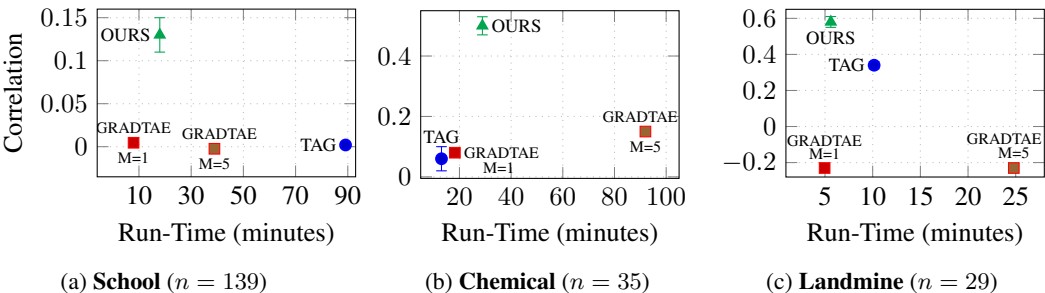

(a) **School** ($n = 139$)  (b) **Chemical** ($n = 35$)  (c) **Landmine** ($n = 29$)

Figure 3: Correlation between the predicted and actual MTL gains ($\pm$ std.) and runtime of various prediction methods. For each benchmark dataset, our method is trained with 25% of all possible pairs ($|\mathcal{G}_{\text{train}}| = 0.25\binom{n}{2}$).

Overall, these results demonstrate that our framework provides a favorable balance between runtime and predictive accuracy.

### 4.3.3 PRACTICAL UTILITY IN DOWNSTREAM SELECTION OF TASK GROUPS

To evaluate the practical utility of our predicted pairwise MTL gains, we use them to select task groups. While exhaustive or branch-and-bound search could, in principle, identify globally optimal task groups (i.e., maximizing predicted gains), these methods are infeasible for our large task sets, where the number of candidate groups grows combinatorially (e.g., tens of thousands per benchmark). This motivates the use of scalable search heuristics: beam search and SDP-based clustering.

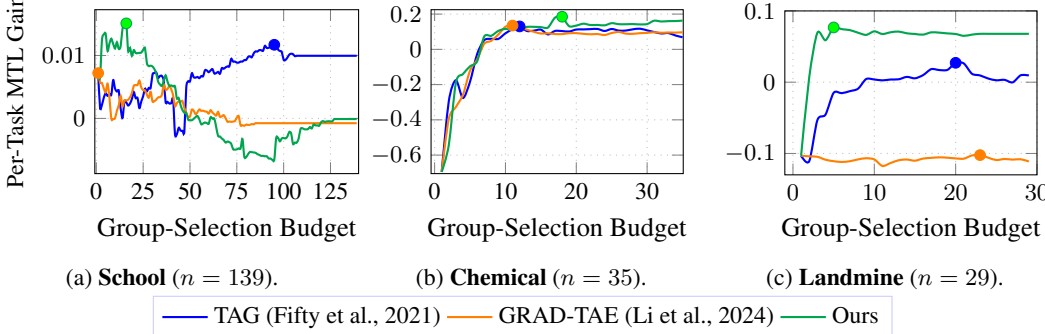

(a) **School** ($n = 139$).  (b) **Chemical** ($n = 35$).  (c) **Landmine** ($n = 29$).

—— TAG (Fifty et al., 2021) —— GRAD-TAE (Li et al., 2024) —— Ours

Figure 4: Comparison between prediction methods (TAG, GRAD-TAE, and ours) when predictions are used for group selection with beam search, evaluated in terms of the average per-task MTL gain of the selected groups.

**Beam Search**  We adopt the beam-search heuristic used by MTGNet (Song et al., 2022), which explores multiple promising groups (i.e., beams) in parallel instead of greedily committing to a single path, with the beam size controlling the tradeoff between thoroughness and cost. To estimate MTL gains for groups of three or more tasks, we aggregate pairwise MTL gain predictions in a group by averaging the predicted gains from all other tasks in a group, similar to Standley et al. (2020) and Fifty et al. (2021) (details in Appendix A.6). At each step of the beam search, the top-scoring candidate groups (i.e., groups with highest predicted MTL gains) are expanded by adding new tasks, scored using predicted gains, and pruned to retain only the best candidates (*School*: 52K, *Chemical*: 26K, *Landmine*: 52K candidate groups). The beam search iterates until the search budget is exhausted. Once the search is finished, we perform MTL training to obtain ground-truth gains for the selected task groups. Figure 4 shows that our prediction approach consistently leads to high-performing groups in terms of MTL gains, outperforming baseline prediction approaches across all benchmarks datasets, despite the modest cost of training our pairwise MTL gain predictor.

**SDP-Based Clustering**  We also apply semidefinite programming (SDP) for task clustering, which we adopt from Li et al. (2024). A square affinity matrix is constructed from predicted pairwise MTL gains (using our approach) or from task-affinity measures (using TAG (Fifty et al., 2021) or GRAD-TAE (Li et al., 2024)). The SDP-based grouping method produces a continuous matrix maximizing

Table 3: Comparison between prediction methods when predictions are used for group selection with SDP, evaluated in terms of total task loss with the selected groups (single-task learning (STL) included for reference).

| | | Total Loss ($\pm$ Std. Dev.) with Optimal Number of Groups $k$ | | | | | |
| | | TAG (Fifty et al., 2021) | | GRAD-TAE (Li et al., 2024) | | Ours | |
| Dataset | STL Loss | Loss | $k$ | Loss | $k$ | Loss | $k$ |
|---|---|---|---|---|---|---|---|
| **School** | 131.47 | $130.30 \pm 0.20$ | 3 | $130.74 \pm 0.30$ | 4 | $\mathbf{130.19} \pm 0.21$ | 3 |
| **Chemical** | 22.98 | $21.64 \pm 0.19$ | 6 | $\mathbf{20.05} \pm 0.66$ | 15 | $20.73 \pm 0.29$ | 2 |
| **Landmine** | 6.09 | $6.25 \pm 0.15$ | 15 | $6.15 \pm 0.16$ | 15 | $\mathbf{5.70} \pm 0.12$ | 4 |

overall similarity under normalization and positivity constraints. Soft assignments are then rounded to form discrete clusters, yielding the desired number of task groups. We have experimented with various numbers of clusters for group selection and report the total loss of all tasks with the optimal number in Table 3 (lower total loss is better). Across all three benchmarks, SDP-based grouping achieves competitive performance with our predicted affinities compared to TAG and GRAD-TAE, outperforming these baselines on *School* and *Landmine* and remaining competitive on *Chemical*.

## 5 RELATED WORK

Understanding task relationships to maximize positive transfer and minimize interference is a core challenge in multi-task learning. Early studies (Caruana, 1997; Argyriou et al., 2008) showed that jointly training tasks with shared structure can improve generalization. Subsequent research has examined task relatedness more broadly, including zero-shot transfer (Pal & Balasubramanian, 2019), representation learning (Dwivedi & Roig, 2019), and information-theoretic perspectives (Achille et al., 2021; Zhuang et al., 2020). Surveys (Zhang & Yang, 2021; Ruder, 2017) provide overviews of task relationships modeling strategies, including shared feature representations, low-rank parameterizations, and task clustering.

Foundational theoretical work of Ben-David & Borbely (2008) formalizes task relatedness through similarity of data-generating distributions, providing generalization guarantees but leaving practical estimation in high dimensions unresolved. Other studies, such as Bingel & Søgaard (2017), analyze task characteristics that correlate with transferability in specific domains (NLP) but do not directly predict pairwise MTL gains. More recent methods shift focus on estimating task-affinities to group tasks prior to MTL training. Many rely on fully data-driven affinity estimation: for instance, Standley et al. (2020) train all task pairs to measure gains, which are then aggregated to predict the benefits of larger groups; MTGNet (Song et al., 2022) and Linear Surrogate models (Li et al., 2023) follow a similar paradigm, requiring extensive MTL training to generalize to unseen groups. Alternatively, some approaches estimate task relatedness via shared feature extractor for all tasks (Shiri & Sun, 2022), or gradient similarity (Yu et al., 2020) or by measuring the effect of gradient updates on each task during MTL training (Fifty et al., 2021), which involves $n$ additional forward/backward passes per step. GRAD-TAE (Li et al., 2024) instead fine-tunes hyperparameters on random groups to infer performance. Though effective, these methods remain computationally expensive due to repeated MTL training or multiple gradient computations and do not scale well to large task collections.

In contrast, we propose a feature-based approach for predicting pairwise MTL gains with minimal training cost. Using statistical, distributional, and representation-level features, our approach enables scalable task-group selection and operationalizes the intuition that tasks with similar data distributions are more likely to benefit from joint training.

## 6 CONCLUSION

We introduced a framework for predicting pairwise MTL gains for tabular datasets accurately and efficiently based on static features of task data. By quantifying divergence in task data through informative task-relation features, our method identifies task affinities with superior accuracy and minimal supervision. Experiments across multiple benchmarks show that it outperforms baselines in terms of predictive accuracy while substantially reducing runtime, demonstrating the potential of feature-driven affinity prediction for scalable and automated task-grouping for MTL.

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

# A APPENDIX

## A.1 EXPERIMENTAL DETAILS

**Dataset Preparation and Statistics:** To ensure fair comparisons in MTL performance, we equalize task dataset sizes, $D_t$ across benchmarks. Since tasks vary in training set size (Table 1), we repeat samples so that each task matches the maximum size, preventing biased updates and improving generalization. For the *Chemical* dataset, we additionally balance positive and negative samples (Jacob et al., 2008).

**Summary of Dataset Statistics:** To better understand the differences in task behavior, we also investigate label variance, standard deviation, and within-task input distance for the benchmarks and summarize the key statistical properties in Table 4. The *School* dataset shows moderate label variability and a diverse input space. *Chemical* dataset exhibits uniform label difficulty with tightly clustered features, while *Landmine* has low label spread and moderately diverse inputs. These characteristics help contextualize differences in task behavior and model performance across datasets.

Table 4: Summery of dataset statistics for pairwise MTL gain prediction study using feature based on task characteristics: label variability and within-task input distance for tasks from the three benchmark datasets.

| Dataset | Variance ($\sigma^2$) (Labels) | Std Dev($\sigma$) (Labels) | Within-Task Distance | Observation |
|---|---|---|---|---|
| **School** | $0.88 \pm 0.29$ | $0.93 \pm 0.15$ | $1.52 \pm 0.24$ | Moderate label variability, diverse input space |
| **Chemical** | $0.25$ | $0.50$ | $0.09 \pm 0.02$ | **Uniform** label difficulty, tightly clustered features |
| **Landmine** | $0.06 \pm 0.01$ | $0.24 \pm 0.03$ | $1.13 \pm 0.09$ | Low label spread, input moderately diverse |

**Model Architecture:** In our experiments with single-task and multi-task learning on the three benchmark datasets, we employ feed-forward neural network (NN) models. Note that the specific MTL architectures are not a vital component to our contribution. In fact, *our goal is orthogonal to the performance of specific MTL methods, and our pairwise MTL gain prediction approach can be applied to and potentially benefit a wide range of MTL methods*. We conduct a random neural architecture search on randomly chosen task groups from each of the three benchmark datasets to select the MTL architectures. We initialize the randomized architecture search with separate task-specific layers and some shared layers among all the tasks. We explore variations in the number of hidden layers, number of neurons per layer, and learning rates to minimize overall loss across the tasks. We finally select the architecture that balances predictive accuracy and model complexity. The best architecture discovered during the search for each benchmark is adopted as the final MTL architecture.

Table 5: Training configuration for benchmark datasets.

| Dataset | Shared Layers | Neurons per Layer | Learning Rate $\alpha$ | Batch Size | Other Settings |
|---|---|---|---|---|---|
| **School** | 3 | [20, 10, 32] | 0.005 | 64 | Num. of epochs = 1000 Early stopping = 50 |
| **Chemical** | 2 | [32, 16] | 0.001 | 264 | Num. of epochs = 1000 Early stopping = 50 |
| **Landmine** | 2 | [64, 32] | 0.001 | 64 | Num. of epochs = 1000 Early stopping = 50 |

**Implementation and Training:** We implement neural network prediction models using Keras with TensorFlow Abadi et al. (2016), minimizing either mean squared error (*School*) or binary cross-entropy (*Chemical* and *Landmine*) with the Adam optimizer Kingma & Ba (2014).

## A.2 ADDITIONAL ANALYSIS

### A.2.1 DISTRIBUTION OF GROUND-TRUTH PAIRWISE MTL GAINS

Before training predictive models, we visualize the empirical distribution of pairwise MTL gains for each dataset. Figure 5 shows histograms of the ground-truth MTL gains for the three benchmark datasets: *School*, *Chemical*, and *Landmine*. To remove extreme outliers and better visualize the core structure, we clip the values between the 0.5th and 99.5th percentiles of each dataset prior to plotting.

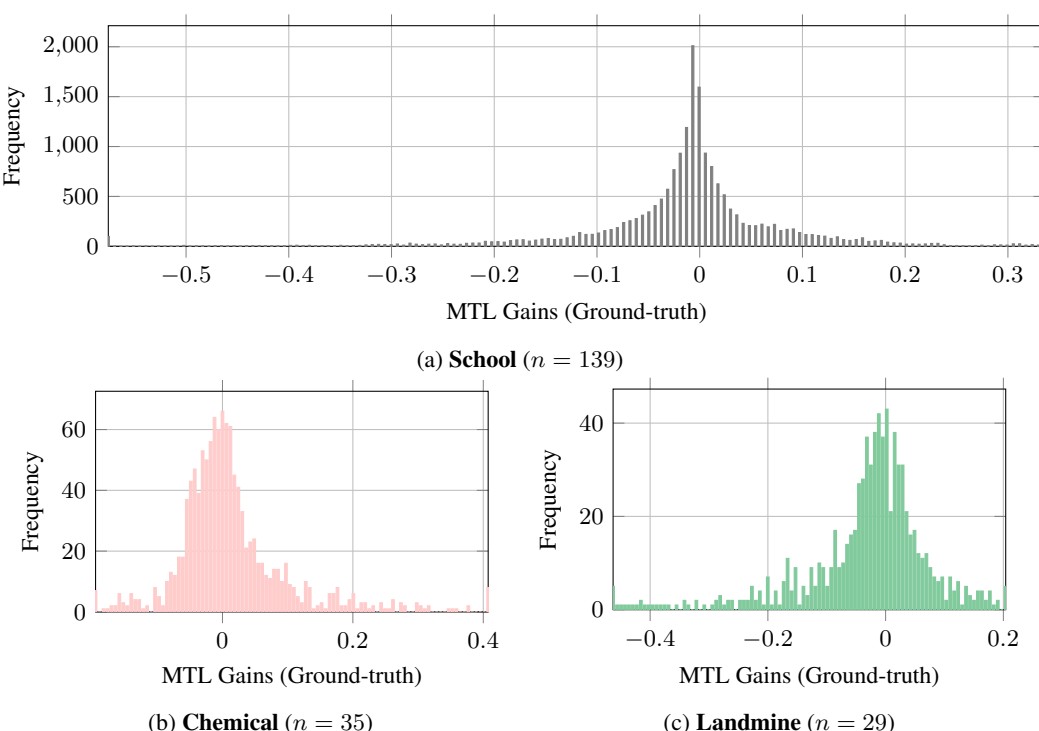

(a) **School** ($n = 139$)

(b) **Chemical** ($n = 35$)

(c) **Landmine** ($n = 29$)

Figure 5: Histograms of Pairwise MTL Gains (Ground-Truth Values) for School, Chemical, and Landmine datasets

Across all datasets, we observe a mix of positive and negative MTL gains:

- *School*: Among 19,182 values (2 gains per 9,591 pairs), 44% are positive (8,422) and 56% negative (10,760), suggesting a slight dominance of negative transfer.
- *Chemical*: Distribution is mildly right-skewed, with 614 positive and 576 negative gains, consistent with the dataset's uniform task difficulty.
- *Landmine*: 344 positive vs. 468 negative gains, indicating a relatively balanced distribution with strong task incompatibilities in some pairs.

These observed gain distributions motivate the need for a robust predictive model that can distinguish between helpful and harmful task pairings before actual training.

### A.2.2 CORRELATION ANALYSIS BETWEEN TASK-RELATION FEATURES AND MTL GAINS

We analyze the Pearson correlations between individual features and observed pairwise MTL gains across *School*, *Chemical*, and *Landmine* datasets. In *School*, correlations are generally weak (all below 0.1 in magnitude), consistent with its moderate label variability and relatively homogeneous task structure—no single feature strongly explains transfer patterns. In contrast, *Chemical* and *Landmine* exhibit more distinctive trends:

- **Data Quantity and Balance:** In *Chemical*, size-driven features such as `Data-Size` (0.45) and `Data-Ratio` (0.16) show moderate positive correlations with gains, sug-

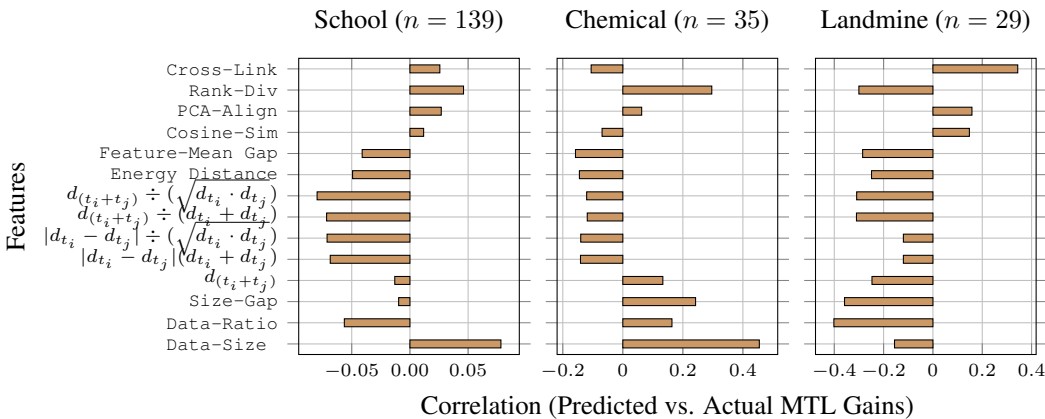

Figure 6: Correlation between task-characteristic features and actual MTL gains across three benchmark datasets.

gesting that task pairs with balanced and sufficiently large data are more likely to bene-fit from MTL. In contrast, these features are weak or negative in *Landmine* ($-0.40$ for `Data-Ratio`), reflecting domain heterogeneity and noise sensitivity.

- **Distance and Distribution Properties:** Distance-based features exhibit consistent nega-tive correlations in *Landmine*, with values as low as $-0.31$ (`normalized inter-task distance`, $d_{(t_i+t_j)} \div (d_{t_i} + d_{t_j})$), indicating that greater dissimilarity in input struc-ture often predicts reduced transferability. `Energy Distance` and `Feature-Mean Gap` also align negatively across datasets, reinforcing the role of distributional mismatch in driving negative transfer.

- **Representation and Graph Features:** Representation-based features such as `Cosine-Sim` and `PCA-Align` have weaker, inconsistent correlations, though they are positive in *Landmine*. The graph-inspired `Cross-Link` feature shows strong positive correlation in *Landmine* ($0.34$) but negligible signal in *School* and *Chemical*.

Overall, these results suggest that while no single feature is universally predictive, dataset bal-ance and distributional divergence are among the most reliable indicators—supporting our use of a learned model that can combine multiple signals.

### A.3 MODEL VARIANTS FOR MTL GAIN PREDICTION: LINEAR VS QUADRATIC REGRESSION

To further examine the effectiveness of our predictive framework, we conducted supplementary experiments comparing alternative modeling choices. Table 6 reports results for two variants trained on the final feature subset: a standard linear regression model and a quadratic regression model with ridge regularization. Performance is measured using $R^2$ scores and Pearson correlation between predicted and ground-truth MTL gains. Each model is trained on $25\%$ of randomly selected task pairs, and results are averaged over multiple random subsets.

Table 6: Comparison of predictive accuracy across model variants for **pairwise MTL gain prediction**. Re-ported are mean $\pm$ std of $R^2$ scores and Pearson correlations between predicted and ground-truth gains.

| Dataset | Feature-Based Prediction Approach | | | |
| --- | --- | --- | --- | --- |
| | Linear Regression | | Quadratic Regression | |
| | $R^2$ | Correlation | $R^2$ | Correlation |
| **School** | $0.01\pm 0.00$ | $0.10\pm 0.01$ | $0.02\pm 0.01$ | $0.13\pm 0.03$ |
| **Chemical** | $0.20\pm 0.04$ | $0.45\pm 0.05$ | $0.22\pm 0.05$ | $0.50\pm 0.03$ |
| **Landmine** | $0.30\pm 0.02$ | $0.56\pm 0.01$ | $0.34\pm 0.03$ | $0.58\pm 0.03$ |

Across all three benchmarks, the quadratic model consistently outperforms its linear counterpart, with the largest gains observed in the *Chemical* and *Landmine* datasets. These findings validate

both our feature selection and modeling strategy, demonstrating that a compact, intuitive feature set combined with a regularized quadratic regression model can effectively approximate pairwise MTL gains.

## A.4 RUNTIME-DETAILS OF DIFFERENT METHODS

Table 7 compares the runtime and predictive quality (correlation with observed MTL gains) of our proposed pairwise gain predictor against two strong baselines, TAG (Fifty et al., 2021) and Grad-TAE (Li et al., 2024), across the three benchmark datasets. The results highlight two key advantages of our approach:

Table 7: Correlation and Runtime between predicted pairwise affinity and actual pairwise MTL gains for different methods. Time for TAG includes a baseline MTL training with inter-task affinity estimation throughout the training. Time for Grad-TAE includes M full Baseline MTL training, plus O(n) gradient evaluations and solving logistic regression m times where m indicates the number of random subsets selected.

| Dataset | TAG | | GRAD-TAE | | | | Ours ($|\mathcal{G}_{\text{train}}| = 0.25 * \binom{n}{2}$) | |
| | Time | Correlation | Time | M=1 Correlation. | Time | M=5 Correlation. | Time | Correlation |
|---|---|---|---|---|---|---|---|---|
| **School** | 89.14 | +0.002 ± 0.00 | 7.78 | +0.005 ± 0.00 | 38.91 | -0.002 ± 0.00 | 17.98 | **+0.13 ± 0.02** |
| **Chemical** | 13.05 | +0.06 ± 0.04 | 18.23 | +0.08 ± 0.01 | 91.17 | +0.15 ± 0.00 | 28.88 | **+0.52 ± 0.02** |
| **Landmine** | 10.15 | +0.34 ± 0.02 | 4.96 | -0.23 ± 0.01 | 24.80 | -0.23 ± 0.01 | 5.61 | **+0.58 ± 0.05** |

First, our method achieves substantially higher predictive accuracy across all datasets. For example, it reaches a correlation of +0.52 on *Chemical* and +0.58 on *Landmine*, outperforming TAG by large margins (+0.06 and +0.34, respectively) and significantly improving over Grad-TAE, which fails to produce meaningful correlation and can even result in negative values (e.g., −0.23 on *Landmine*).

Second, our method provides a favorable trade-off between runtime and prediction quality. While TAG requires full MTL training with inter-task affinity computation throughout (e.g., 89.14 minutes on *School*), and Grad-TAE with $M = 5$ scales linearly with the number of baseline trainings (reaching 91.17 minutes on *Chemical*), our approach only requires partial pairwise training (25% of all pairs) and converges in markedly less time (e.g., 17.98 minutes on *School* and 5.61 minutes on *Landmine*).

Overall, these demonstrate that our predictor not only delivers the strongest correlation with true pairwise gains but also does so with lower or competitive runtime, making it a more scalable and practical choice for large-scale multi-task learning settings.

**Data-Efficiency of Our Approach** Additionally, Figure 7 shows how prediction performance of our proposed approach improves as we increase the number of training pairs used for learning the gain predictor. Across all three datasets—*School*, *Chemical*, and *Landmine*—we observe a clear upward trend: as more pairs are incorporated, the correlation between predicted and observed MTL gains consistently increases, with diminishing returns as the training set approaches 50% of all available pairs.

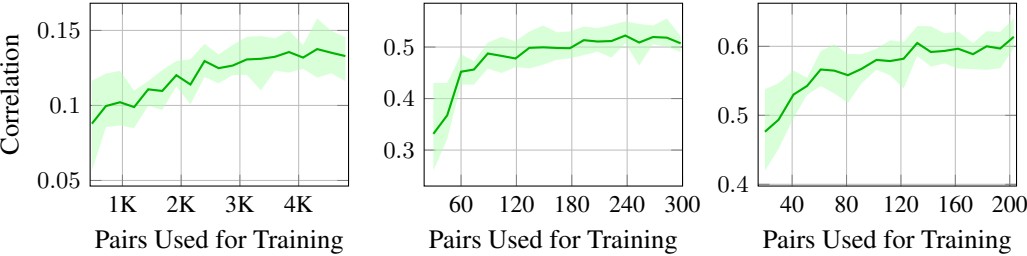

Figure 7: Prediction performance of our approach (correlation between predicted and actual MTL gains) vs. number of pairs used for training the predictor ($|\mathcal{G}_{\text{train}}|$), with shaded bands indicating the interquartile range (25th–75th percentile). Training set $\mathcal{G}_{\text{train}}$ comprises 10–50% of all task pairs.

This result demonstrates that the proposed predictor is data-efficient: even with a small fraction of available task pairs (5–10%), it achieves meaningful correlation (e.g., 0.10 for *School*, 0.45 for

*Chemical*, 0.48 for *Landmine*). As training coverage grows, prediction quality steadily improves, eventually saturating around 30–40% of pairs. These findings highlight the benefit of collecting more labeled pairs for training but also show that reliable performance can be achieved without exhaustively evaluating every pair. This makes our approach practical for real-world MTL settings, where the total number of task pairs grows quadratically with the number of tasks and exhaustive evaluation is infeasible.

## A.5 COMPARISON WITH OTHER BASELINE

### A.5.1 LINEAR SURROGATE MODELS FOR PAIRWISE GAIN PREDICTION

To compare our approach with additional baselines, we implemented a **linear surrogate model** (Li et al., 2023) that takes as input a binary task-presence vector (indicating which tasks are included in the group) and directly regresses on the observed multi-task learning (MTL) gains. The model is evaluated on only pairs, allowing for a fair comparison with our feature-based pairwise MTL gain predictor.

Table 8: Performance summary of **Linear Surrogate** (Li et al., 2023) baseline: Overall metrics refer to the model evaluated on predictions for all tasks jointly, while per-task metrics report the mean and standard deviation across predictions for individual tasks.

| Dataset | Training Groups | Runtime (minutes) (Prepare $\mathcal{G}_{\text{train}}$) | Overall (for all tasks) $R^2$ | Corr. | Avg. Per-task Metrics Corr ($\pm$ std) | $R^2$ ($\pm$ std) |
|---|---|---|---|---|---|---|
| **School** | 50 | 25.6 | 0.165 | 0.42 | $0.003 \pm 0.08$ | $-0.86 \pm 3.435$ |
| | 100 | 51.2 | 0.167 | 0.42 | $0.007 \pm 0.08$ | $-0.80 \pm 3.508$ |
| | 200 | 102.4 | 0.152 | 0.40 | $0.014 \pm 0.08$ | $-0.83 \pm 3.728$ |
| **Chemical** | 30 | 161.7 | -3.718 | 0.14 | $0.02 \pm 0.18$ | $-5.54 \pm 7.233$ |
| | 100 | 539.0 | very neg | 0.12 | $0.09 \pm 0.20$ | $-1.2\text{e}30 \pm 2.9\text{e}30$ |
| | 200 | 1078.0 | very neg | 0.04 | $0.16 \pm 0.25$ | $-2.1\text{e}29 \pm 1.2\text{e}30$ |
| **Landmine** | 30 | 18.5 | -1.5 | 0.30 | $0.09 \pm 0.20$ | $-1.74 \pm 1.931$ |
| | 100 | 61.7 | -9.9e26 | 0.05 | $0.17 \pm 0.21$ | $-1.8\text{e}27 \pm 9.6\text{e}27$ |
| | 200 | 123.4 | -9.8e28 | -0.07 | $0.24 \pm 0.20$ | $-6.1\text{e}29 \pm 1.7\text{e}30$ |

Table 8 summarizes the model's performance across different training set sizes. Per-task metrics report the mean and standard deviation for each task individually, whereas overall metrics aggregate predictions across all tasks. Differences between these metrics arise because tasks with very low variance or poorly predicted outcomes can disproportionately affect per-task $R^2$, while the overall metric smooths these effects. Similarly, per-task correlations are specific to a single task, whereas overall correlation is computed across a mixture of tasks with different scales, which can "average out" and lead to higher values.

For the *School* dataset, the linear surrogate achieves slightly higher correlation and $R^2$ than our feature-based predictor, although the cost in terms of runtime is much higher. However, this performance does not generalize: for *Chemical* and *Landmine*, the surrogate model drastically overfits, producing near-zero or negative correlation values and extremely large (sometimes negative) $R^2$ scores — clear indications of instability and non-identifiability. This behavior demonstrates that the high-dimensional 0-1 input encoding does not capture meaningful similarity information for pairwise transfer prediction, rendering the problem **non-identifiable under a purely linear mapping**.

These results emphasize the necessity of feature engineering and inductive bias in predicting MTL gains: without carefully designed inter-task features, as used in our method, the learning problem becomes ill-posed and prone to memorization rather than generalization.

## A.6 DOWNSTREAM GROUP-SELECTION USING PREDICTED PAIRWISE MTL GAINS

**Beam Search for Task Group Selection.** Beam search is a heuristic strategy that explores several promising partial solutions in parallel instead of committing to a single greedy path. The application of beam search follows two steps: we first aggregate the pairwise predictions to estimate group-level MTL gains for sets of three or more tasks. To estimate the potential performance of larger task groups (with three or more tasks), we build upon previous work by Standley et al. (2020) and Fifty

et al. (2021), which propose Higher-Order Affinity (HOA) aggregation techniques. In particular, for a given task group $G$ and a task $t_i \in G$, we compute the group-to-task MTL gain by averaging the pairwise predicted affinities from all other tasks in the group to $t_i$:

$$\hat{\text{gain}}_{G \to t_i} = \frac{1}{|G| - 1} \sum_{t_j \in G, j \neq i} \hat{\text{gain}}_{t_j \to t_i} \tag{15}$$

This yields a group-level prediction vector $\hat{\boldsymbol{gain}}_G = \{\hat{gain}_{G \to t} : t \in G\}$, which we interpret as the predicted MTL gain values for the task group $G$. Once we have the group-level predictions, we apply the beam search algorithm on the candidate set of task-groups. At each step, the beam search keeps the top-scoring candidate groups (the beam), expands them by adding new tasks, scores the resulting groups using predicted gains, and prunes back to retain only the best candidates. This process repeats until the search budget is exhausted, after which the group with the highest estimated total MTL gain is selected. Beam size controls the tradeoff between accuracy and computational cost, allowing flexible adjustment. We adopt the grouping selection algorithm from MTGNet (Song et al., 2022), and refer readers to the original paper for a more detailed explanation.

**Spectral SDP-Based Clustering for Task Grouping.** To form task groups, we first construct a square affinity matrix that captures pairwise relationships between all tasks. The $n \times n$ affinity matrix can be obtained using predictions made by our approach, or from any alternative task-similarity method such as TAG (Fifty et al., 2021) or GRADTAE (Li et al., 2024).

We then solve a semidefinite programming (SDP) relaxation that seeks a matrix representation of task assignments which maximizes overall similarity while satisfying normalization and positivity constraints. The solution to this optimization problem is a continuous matrix indicating soft group memberships. Finally, we convert this soft solution into discrete task groups by applying a rounding procedure based on a similarity threshold. This produces exactly the desired number of task clusters. The group-selection procedure is adopted from prior work on SDP-based clustering (Li et al., 2024), and alternative solvers or rounding strategies could be used without changing the overall framework.

