# OpenReview forum: "Identification of Task Affinity for Multi-Task Learning based on Divergence of Task Data"
_ICLR.cc/2026/Conference — Submitted to ICLR 2026_

### Official Review · Reviewer_4wip · 2025-10-26

**Soundness:** 2
**Presentation:** 2
**Contribution:** 2
**Rating:** 4
**Confidence:** 4

**Summary:**

The paper proposes a feature-based approach for predicting pairwise task affinity in multi-task learning (MTL) for tabular datasets.

**Strengths:**

The paper includes evaluation on three different benchmark datasets with different characteristics, and the experiments cover multiple aspects including prediction accuracy, computational cost, and downstream task grouping performance.

**Weaknesses:**

1. The biggest limitation is that this approach only works for tabular datasets with shared input dimensions. The authors acknowledge this (line 64-: "We assume a common input dimension p across all tasks"), but this severely limits applicability. Most interesting MTL problems in computer vision or NLP don't have this property. The paper should discuss more clearly when this assumption holds and provide examples beyond the three benchmarks tested.
2. The paper is quite empirical. While the hypothesis that "tasks with more similar data distributions benefit from joint training" is intuitive, there's little theoretical justification for why these specific features should predict MTL gains. Some analysis connecting feature values to properties that affect gradient-based optimization or representation learning would strengthen the work.
3. Some parts are repetitive (e.g., the motivation for feature-based prediction is stated multiple times). Some experimental details are unclear (e.g., how exactly is the train/test split done? Are results averaged over multiple random splits?)
4. Table 4: "Std Dev(σ)", why not just σ?
5. In Table 3, why does the optimal number of groups k vary so much (2-15)? How sensitive is the final performance to choosing k?
6. Can you provide more intuition or theoretical justification for why these specific features should predict MTL gains? For instance, why should Energy Distance be particularly predictive?

**Questions:**

See weaknesses.

---

> ### Author Response · Authors · 2025-11-27
>
> ## W1: Tabular Data vs. Other Modalities
>
> The goal of our work is to introduce an approach that is **tailored to tabular data, achieving significantly better efficiency and accuracy than existing, modality-agnostic approaches**. While deep learning research frequently prioritizes unstructured data (e.g., images), tabular data remains the dominant modality for many high-stakes industry applications, such as fraud detection across diverse financial transaction logs or diagnostic modeling within clinical databases. Unlike vision or language, tabular tasks lack universal structural priors (such as spatial locality or sequential dependencies), necessitating a tailored approach that explicitly leverages statistical properties to identify task affinities in heterogeneous feature spaces.
>
> To extend our approach to unstructured data (e.g., images), we could replace the raw tabular data features with embeddings from pre-trained feature-extraction models (e.g., ResNet or ViT). We could then compute our proposed distributional statistics, such as Energy Distance and Feature-Mean Gap, directly and efficiently in this embedding space. This would allow us to predict affinity without determining task gradients or training domain-specific extractors (like VAEs), preserving the computational efficiency of our approach. However, this is outside the scope of this paper.
>
> ## W2 and W6: Theoretical Justification
>
>
> We agree that while our primary contribution is a practical, scalable framework, grounding these empirical features in theoretical principles is important. While we avoided heavy theoretical derivation to focus on computability and scalability, **our feature design was indeed guided by theoretical principles of domain adaptation and multi-task optimization**; we will add a discussion to our paper, which will explicitly connect our statistical features to these principles.
>
> First, existing theory in domain adaptation (e.g., Ben-David & Borbely, 2008) establishes that the generalization error of a model trained on a source domain and applied to a target domain is bounded by the divergence between their input distributions. Our Energy Distance and Feature-Mean Gap features serve as computationally efficient proxies for this divergence. Empirically, we find that lower values of these metrics correlate with positive transfer, which is consistent with the theoretical expectation that higher distributional similarity reduces the bound on generalization error and facilitates shared learning.
>
> Furthermore, our structural features directly relate to the geometry of the input space and its influence on “gradient interference” during gradient-based optimization. With hard parameter sharing, negative transfer often occurs when tasks compete for model capacity, leading to conflicting gradient updates where an update that improves one task harms another. Features such as Cross-Link Ratio and Normalized Distances quantify the topological overlap between task data; disjoint data distributions compel a shared encoder to map distinct input regions to potentially divergent representations, which can lead to conflicting gradients during backpropagation. Finally, features like PCA-Alignment directly assess the assumption that tasks share a low-dimensional subspace, which is a prerequisite for the efficacy of shared representation learning. By measuring these properties statically, we are efficiently estimating the likelihood that the interference between tasks will be positive.
>
> ## W3: Presentation
>
> We appreciate the feedback regarding the presentation and will streamline the text to reduce redundancy. We will also clarify the experimental details, such as the usage of multiple random splits.
>
> ## W4: Notation in Table 4
>
> Thank you for pointing this out. We agree that the notation “Std Dev($\sigma$)” (and “Variance ($\sigma^2$)”) is redundant. We will revise the column header to simply “$\sigma$” (and “$\sigma^2$”).

---

> > ### Author Response · Authors · 2025-11-27
> >
> > ## W5: Optimal Number of Groups in Table 3
> >
> > The variation in the optimal number of groups ($k$) is not an artifact of instability, but rather reflects: (1) the intrinsic task relationships within each dataset, and (2) the effectiveness of each prediction method in identifying these underlying structures. Datasets with high homogeneity (e.g., *School*) benefit from broad sharing in fewer task groups (e.g., $k=3$ or $4$), while other datasets (e.g., *Landmine*) exhibit strong pairwise incompatibilities that necessitate finer-grained task grouping to avoid negative transfer. Notably, while the baselines predictors fragmented *Chemical* and *Landmine* tasks significantly (up to $k=15$), our feature-based predictor identified a more efficient, consolidated structure ($k=2$ and $4$). This demonstrates that our approach was more effective at finding compatible sub-structures, while the baseline approaches failed to identify opportunities for sharing.
> >
> > Regarding the sensitivity of the results, it is important to note that $k$ is chosen by the clustering algorithm based on the predicted affinity matrix, rather than being a manually tuned hyperparameter. We will revise the text to discuss the optimal number of groups and to clarify this distinction regarding the algorithmic selection of $k$.

---

### Official Review · Reviewer_ahwk · 2025-10-31

**Soundness:** 3
**Presentation:** 3
**Contribution:** 2
**Rating:** 6
**Confidence:** 4

**Summary:**

This paper proposes an approach to predict task affinity in multitask learning (how beneficial it is to train two tasks together), based on static, precomputed features of the task datasets, without requiring expensive joint training for all task pairs.
- Instead of treating task relationships as a black box, the paper quantifies statistical and structural similarities between tasks using easily computed dataset-level metrics. These include measures of dataset size, input-space distances, distributional divergence (e.g., energy distance, feature mean gaps), and representation similarity (e.g., cosine similarity, PCA alignment).
- The authors construct pairwise feature vectors from these metrics and train a quadratic regression model to predict MTL gains, defined as the relative improvement in task performance when trained jointly versus independently. Crucially, the model is trained only on a small subset of task pairs with known ground-truth MTL gains.

The framework was tested on three standard tabular MTL benchmarks, School, Chemical, and Landmine datasets. The proposed model outperformed prior task-affinity estimation methods, such as TAG (gradient-based affinity estimation) and GRAD-TAE (gradient-projection model), in both prediction accuracy and computational efficiency. For example, the proposed method achieved correlations up to 0.58 with true MTL gains on the Landmine dataset, while requiring only a fraction of the training time needed by baselines. Moreover, when applied to task-group selection using beam search and semidefinite programming clustering, the predicted affinities led to superior task groupings with lower total loss compared to alternatives.

**Strengths:**

- This work contributes a scalable and interpretable method for predicting task affinities using simple dataset-derived features, reducing the computation for MTL training.

- It provides a practical pathway to automatic task grouping, particularly for tabular data with many tasks. The authors’ findings support the hypothesis that tasks with more similar data distributions yield more positive transfer, offering both theoretical and empirical validation.

- The approach advances MTL research by improving efficiency and accuracy in task affinity prediction, enabling large-scale applications of multi-task learning.

**Weaknesses:**

- The framework focuses on tabular data and extracts dataset features based on tabular features. It would be better to understand how this framework can be applied in a generic setting, for example, in the scenarios of deep neural networks & image/text datasets.

- The method predicts pairwise task affinity. Would it be possible to extend the framework to predicting higher-order task affinities, such as the multitask learning results when training on more than two tasks?

- Is there any theoretical analysis on the prediction accuracy of using a linear model on the extracted features?

**Questions:**

Please see the weaknesses.

---

> ### Author Response · Authors · 2025-11-27
>
> ## W1: Tabular Data vs. Other Modalities
>
> The goal of our work is to introduce an approach that is **tailored to tabular data, achieving significantly better efficiency and accuracy than existing, modality-agnostic approaches**. While deep learning research frequently prioritizes unstructured data (e.g., images), tabular data remains the dominant modality for many high-stakes industry applications, such as fraud detection across diverse financial transaction logs or diagnostic modeling within clinical databases. Unlike vision or language, tabular tasks lack universal structural priors (such as spatial locality or sequential dependencies), necessitating a tailored approach that explicitly leverages statistical properties to identify task affinities in heterogeneous feature spaces.
>
> To extend our approach to unstructured data (e.g., images), we could replace the raw tabular data features with embeddings from pre-trained feature-extraction models (e.g., ResNet or ViT). We could then compute our proposed distributional statistics, such as Energy Distance and Feature-Mean Gap, directly and efficiently in this embedding space. This would allow us to predict affinity without determining task gradients or training domain-specific extractors (like VAEs), preserving the computational efficiency of our approach. However, this is outside the scope of this paper.
>
> ## W2: Higher-Order Task Affinities
>
> The reviewer raises a very good question. Yes, it is possible to extend our pairwise framework to higher-order task groups. In fact, **we have already used an aggregation strategy for some of the task-grouping experiments** in Section 4.3.3.
>
> As detailed in Appendix A.6 (Equation 15) and Section 4.3.3, we estimate the Multi-Task Learning (MTL) gain for a task within a larger group (size > 2) by averaging the predicted pairwise affinities between that task and every other task in the group. This strategy follows the Higher-Order Affinity (HOA) aggregation technique proposed by Standley et al. (2020), which is also followed by TAG (Fifty et al., 2021). We utilized this higher-order estimation strategy for the task-grouping experiments with *beam search* (Figure 4). As alternative to estimating higher-order task affinities, we also provided experimental results on grouping tasks with *SDP-based clustering* (Table 3); this method relies directly on pairwise values, without requiring explicit higher-order affinity predictions.
>
> ## W3: Linear Model for Prediction
>
> In this work, we focused on the empirical validation and practical utility of our framework for scalable task grouping, as deriving formal bounds that connect static dataset statistics to complex MTL training dynamics remains a significant open challenge. We provide strong empirical evidence regarding the behavior of linear and non-linear models on our extracted features. Our experiments in Appendix A.3 show that a **quadratic model consistently yields higher accuracy than a linear one** (e.g., higher $R^2$ and correlation). This demonstrates that while a linear approximation provides a reasonable baseline, the mapping between our statistical features and MTL gain involves higher-order interactions. Additionally, our analysis in Appendix A.5.1 demonstrates that without our proposed features, a linear model becomes non-identifiable and overfits, highlighting that the theoretical value of our work lies in **identifying specific statistical features that make the affinity prediction problem tractable and learnable**, rather than in the complexity of the regressor itself.

---

### Official Review · Reviewer_sUsc · 2025-11-01

**Soundness:** 3
**Presentation:** 2
**Contribution:** 2
**Rating:** 2
**Confidence:** 4

**Summary:**

This paper predict task affinity for multi-task learning using tabular dataset statiscis and static features can characterize the reslationship between the tasks without pair-wise exhasutive joint training as previous works.

**Strengths:**

1. The work avoids the combinatorial cost of training on all task pairs, enabling scalable and inexpensive estimation of pairwise affinities for large task sets.
2. The method efficiently identifies high-performing task groups.

**Weaknesses:**

1. The method is validated only on tabular MTL, so it is unclear whether the findings transfer to vision or NLP. Evaluations on standard vision multi-task benchmarks (e.g., Taskonomy) are needed to establish external validity.

2. Task similarity can evolve during training, but the proposed feature-based metric is essentially static and costly to refresh. Prior work such as Selective Task Group Updates for Multi-Task Optimization [1] and GRAD-TAE [2] can track changing task affinities without exhaustive joint training.

[1] Selective Task Group Updates for Multi-Task Optimization (ICLR 2025)

[2] Scalable Multitask Learning Using Gradient-based Estimation of Task Affinity (ACM SIGCOMM 2024)

3. The estimator uses a hand-picked subset of features. When indicators disagree, the decision rule and its justification are unclear. A more principled aggregation is needed to show that dependence on such statistics is appropriate.

**Questions:**

Please respond to these concerns: external validity beyond tabular data, including results on standard vision MTL benchmarks such as Taskonomy, and how your approach handles non-stationary task relations compared to methods that track changing affinities without exhaustive joint training. Also clarify how conflicting feature signals are resolved, provide justification for relying on simple statistics.

---

> ### Author Response · Authors · 2025-11-27
>
> ## W1 / Q1: Tabular Data vs. Other Modalities
>
> The goal of our work is to introduce an approach that is **tailored to tabular data, achieving significantly better efficiency and accuracy than existing, modality-agnostic approaches**. While deep learning research frequently prioritizes unstructured data (e.g., images), tabular data remains the dominant modality for many high-stakes industry applications, such as fraud detection across diverse financial transaction logs or diagnostic modeling within clinical databases. Unlike vision or language, tabular tasks lack universal structural priors (such as spatial locality or sequential dependencies), necessitating a tailored approach that explicitly leverages statistical properties to identify task affinities in heterogeneous feature spaces.
>
> To extend our approach to unstructured data (e.g., images), we could replace the raw tabular data features with embeddings from pre-trained feature-extraction models (e.g., ResNet or ViT). We could then compute our proposed distributional statistics, such as Energy Distance and Feature-Mean Gap, directly and efficiently in this embedding space. This would allow us to predict affinity without determining task gradients or training domain-specific extractors (like VAEs), preserving the computational efficiency of our approach. However, this is outside the scope of this paper.
>
> ## W2 / Q2: Evolving Task Relations
>
> We would like to clarify that **our primary objective is task grouping, where the decisive metric is the final MTL gain** (reduction in converged loss) rather than the trajectory of transfer during training. While task relations may fluctuate during optimization, our goal is to identify which tasks ultimately benefit from joint training; therefore, intermediate dynamics are less relevant to the grouping decision. Our focus on predicting cumulative affinity aligns with established task-grouping literature, including TAG (Fifty et al., 2021), MTGNet (Song et al., 2022), and GRAD-TAE (Li et al., 2024).
>
> ## W3 / Q3: Conflicting Statistical Features and Predictions
>
> We would like to clarify that our **proposed statistical features are intended as inputs for a predictive model** rather than as standalone direct indicators, and we do not rely on manual heuristics or “hand-picked” decision rules to resolve conflicting signals. Instead, we employ a supervised regression model (Section 3.2) that learns to aggregate these features and resolve conflicting signals by optimizing weights based on ground-truth training data. Regarding feature selection, our subset was chosen through a systematic analysis of predictive power and redundancy (Section 4.2.1, Figure 1), ensuring that we cover key axes of variation such as dataset balance and distributional divergence. Finally, we rely on these simple statistical features specifically to avoid the high computational cost of gradient-based methods, and our results demonstrate that these lightweight features are sufficient to outperform training-intensive baselines in terms of both accuracy and speed (Table 2). (For a discussion of theoretical underpinnings, please see *Reviewer 4wip / W2 and W6: Theoretical Justification*.)

---

### Author Response · Authors · 2025-12-03

To the Area Chair and Reviewers,

We thank the Reviewers (`sUsc`, `ahwk`, `4wip`) for their constructive feedback and for recognizing the novelty of our work, and we thank the Area Chair for managing the review process in these unusual conditions.

There is a consensus among the reviewers regarding the **key strengths** of our paper: introducing a **novel approach for identifying task affinity** for multi-task learning (MTL) that is **computationally inexpensive**, **scalable for large sets of tasks**, and **accurately identifies high-performing task groups**, enabling effective and automated downstream task grouping for MTL. However, the reviewers also raised a number of questions and concerns. We addressed these during the rebuttal phase, but the discussion was closed before the reviewers could respond. Below, we provide a summary of some of the key clarifications that we provided.

## Tabular Data vs. Other Modalities
All three reviewers noted that our proposed approach is limited to tabular data. We would like to emphasize that this is a deliberate choice of scope: our goal was to introduce an approach that is **tailored to tabular data, achieving significantly better efficiency and accuracy than existing, modality-agnostic approaches**. While deep learning research frequently prioritizes unstructured data (e.g., images), tabular data remains the dominant modality for many high-stakes industry applications, such as fraud detection across diverse financial transaction logs or diagnostic modeling within clinical databases. Given the prevalence of tabular data in real-world, industry applications of machine learning, we believe that focusing on tabular data is well motivated. Unlike vision or language, tabular tasks lack universal structural priors (such as spatial locality or sequential dependencies), necessitating a tailored approach that explicitly leverages statistical properties to identify task affinities in heterogeneous feature spaces. We would also like to note that it would be possible to extend our approach to unstructured data (e.g., images), as we discuss in the comments below; however, this is outside the scope of our current work.
## Reviewer sUsc
**Evolving task relations (W2 & Q2):** Our primary objective is task grouping, where the decisive metric is the final MTL gain (reduction in converged loss) rather than the trajectory of transfer during training. While task relations may fluctuate during optimization, our goal is to identify which tasks ultimately benefit from joint training; therefore, intermediate dynamics are less relevant to the grouping decision. Our focus on cumulative affinity aligns with established task-grouping literature, including TAG (Fifty et al., 2021), MTGNet (Song et al., 2022), and Grad-TAE (Li et al., 2024).
**Conflicting Statistical Features and Predictions (W3 & Q3):** Our statistical features are intended as inputs for a predictive model rather than as direct indicators, and we do not rely on manual heuristics or “hand-picked” decision rules to resolve conflicting signals. Instead, we employ a supervised regression model (Section 3.2) that learns to aggregate features and resolve conflicting signals based on ground-truth training data.
## Reviewer 4wip
**Theoretical justification (W2 & W6):** We agree that grounding features in theoretical principles is important. While we avoided heavy theoretical derivation to focus on a practical and scalable framework, **our feature design was indeed guided by theoretical principles of domain adaptation and multi-task optimization**. First, existing theory in domain adaptation (e.g., Ben-David & Borbely, 2008) establishes that the generalization error of a model trained on a source domain and applied to a target domain is bounded by the divergence between their input distributions. Our Energy Distance and Feature-Mean Gap features serve as computationally efficient proxies for this divergence. Empirically, we find that lower values of these metrics correlate with positive transfer, which is consistent with the theoretical expectation. Second, our structural features directly relate to the geometry of the input space and its influence on “gradient interference” during gradient-based optimization. With hard parameter sharing, negative transfer often occurs when tasks compete for model capacity, leading to conflicting gradient updates. Features such as Cross-Link Ratio and Normalized Distances quantify the topological overlap between task data; disjoint data distributions compel a shared encoder to map distinct input regions to potentially divergent representations, which can lead to conflicting gradients. Finally, features like PCA-Alignment directly assess the assumption that tasks share a low-dimensional subspace, which is a prerequisite for the efficacy of shared representation learning. By measuring these properties statically, we efficiently estimate the likelihood that the interference between tasks will be positive.

---

### Meta-Review · Area_Chair_4zcH · 2026-01-07

**Summary:**

In the initial phase, the paper received mixed scores (2, 6, 4), with reviewers raising substantial concerns. The primary issues highlighted were:

Reviewer sUsc: The method is limited to tabular multi-task learning (MTL) and assumes static task relations.

Reviewer ahwk: The approach is restricted to tabular MTL, considers only pairwise task affinities, and lacks theoretical analysis.

Reviewer 4wip: The method is limited to tabular MTL, lacks theoretical analysis, and has some writing issues.

Overall, the reviewers' concerns converge on the following key limitations: restriction to tabular MTL, absence of theoretical analysis, reliance on static task relations, and consideration of only pairwise task affinities.

**Reviewer Concerns:**

All reviewers expressed concerns regarding the model's restriction to tabular multi-task learning (MTL). In my view, the rebuttal does not adequately address this limitation. While the authors argue that their method is intentionally designed for tabular data, this response fails to engage with the reviewers' core request: a demonstration of the method's potential extension to broader MTL settings and other data modalities (e.g., Taskonomy).

**Reviewer Scores:**

Given that the primary concern regarding the method's restriction to tabular MTL remains unaddressed, I think that reviewers will either maintain their original scores (leaning toward rejection) or lower them further.

---

### Decision · Program_Chairs · 2026-01-26

Reject